# Separate Adjustment of Linear and Nonlinear Parameters in Neural Network Training

## Abstract

The paper examines the limitations of the backpropagation error (BPE) method in neural network training, particularly its tendency to converge to suboptimal local minima. Traditional backpropagation-based training often suffers from inefficiencies in high-dimensional and complex optimization landscapes, which limits its effectiveness in deep learning applications. A modified neuron model is proposed, featuring adjustable parameters for nonlinear transformations such as ReLU and SoftPlus, which are adapted independently from connection weights. Unlike conventional models, which rely solely on weight optimization, our approach introduces independent parameter tuning for nonlinear transformations, allowing for more efficient exploration of the loss landscape. Based on vector-matrix analysis, the paper introduces an improved formal neuron model that reduces the likelihood of convergence to local minima far from the globally optimal solution. In the proposed model, the output activity is expressed as the sum of linear activation and its nonlinear transformation. This approach significantly enhances training speed and, in particular, approximation accuracy by introducing tunable parameters into the nonlinear function and optimizing them separately from the adjustment of input connection weights. The proposed model was evaluated on function approximation tasks of varying complexity in two- and three-dimensional spaces. The results demonstrate a 3–10 times reduction in training time and up to three orders of magnitude improvement in accuracy, especially for SoftPlus activation. These findings suggest that the proposed neuron model could be beneficial for deep learning applications requiring high precision and efficient training, such as medical imaging and autonomous systems. Additionally, the results emphasize the potential of vector-matrix analysis in improving neural network training methods, paving the way for further exploration of specialized optimization techniques.

## 1 Introduction

The proposed model of a "separated" formal neuron is based on the idea that splitting the parameters of nonlinear activation functions into individual and customizable components can significantly improve the training process of neural networks and overcome issues related to stagnation and local minima. In this context, various optimization methods and activation function tuning strategies, such as ReLU and SoftPlus, are considered, aiming to enhance the learning performance of deep networks.

### 1.1 SoftPlus and ReLU in Deep Neural Networks

Several studies focus on improving deep neural networks' efficiency by utilizing different activation functions. Prince (2023) discusses the use of the SoftPlus activation function, which enhances the performance of deep neural networks by mitigating the issue of vanishing gradients during backpropagation. SoftPlus demonstrates advantages over ReLU due to its smoothness and non-zero derivative properties, making it particularly beneficial in phoneme recognition tasks. Unlike ReLU, which has abrupt transitions, SoftPlus provides smoother boundaries, improving gradient stability and reducing the risk of shallow minima.

## 1.2 Adaptive Learning Methods in Neural Networks

Neural networks leverage adaptive learning strategies to adjust parameters based on data structure and complexity. The study Singh (2023) introduces a criterion for active learning, ensuring that neural networks generalize effectively while remaining robust to initialization settings. This approach is particularly useful for models with individually tunable activation parameters, such as the proposed separated neuron model.

## 1.3 Classification Using Radial Basis Function Networks

The work Marfo & Przybyła-Kasperek (2022) explores radial basis function (RBF) networks for classification, where data is gathered from independent sources. Although not a direct analog of the proposed model, this approach aligns with the concept of separate learning for different parameter sets. Comparative analysis with multi-layer perceptrons Przybyła-Kasperek & Marfo (2024) indicates that RBF networks can reduce error rates and model complexity.

## 1.4 Handling Imbalanced Data Streams

The study Czarnowski (2022) presents the Weighted Ensemble with One-Class Classification and Over-Sampling and Instance Selection (WECOI) method, which addresses the issue of imbalanced data. This method leverages ensemble classifiers and instance selection techniques to balance class distributions in streaming data. This approach may benefit separated neuron training by effectively handling activation parameter differences across diverse data segments.

The proposed separated formal neuron model shares similarities with various contemporary methods and can leverage advances in modified activation functions, active learning, and data processing techniques to improve its performance.

## 2 Description of Activities and Connection Weights Using Vectors and Matrices

All $N^{(l-1)}$ incoming connection weights to neuron $i$ of layer $l$ are described by the weight vector $\boldsymbol{w}_i^l = \{w_{ji}^l\}$ $(i = 0, 1, \ldots, N^l; j = 0, 1, \ldots, N^{(l-1)})$. When different input signals $\boldsymbol{x}$ are fed into the neural network, the activation propagates through the network and determines the values of vector $\boldsymbol{o}^{(l-1)}$—the activity of the elements in the previous (for layer $l$) layer. The vector $\boldsymbol{o}^{(l-1)}$ defines the linear activation $a_i^l$ of neuron $i$ in layer $l$ according to the formula:

$$a_i^l = \boldsymbol{w}_i^l \cdot \boldsymbol{o}^{(l-1)} \tag{1}$$

The magnitude of the (nonlinear) output activity is formed by applying a nonlinear transformation to the scalar value $a_i^l$, resulting in $o_i^l = \varphi(a_i^l)$. For simplicity, we will primarily consider the basic nonlinear transformations:

- **ReLU**: $\varphi(a_i^l) = \max(0, a_i^l)$

- **SoftPlus**: $\varphi(a_i^l) = \log(1 + e^{a_i^l})$

Other nonlinear functions can be used, but for the objectives of this study, ReLU and SoftPlus are the most suitable.

In vector form, the activation of network neurons is expressed as:

$$\boldsymbol{A}^l = \boldsymbol{W}^l \boldsymbol{o}^{(l-1)} \tag{2}$$

$$\mathbf{A}^l = \{a_i^l = \mathbf{W}_i^l \mathbf{O}^{(l-1)}\} \tag{3}$$

$$\boldsymbol{o}^l = \varphi(\boldsymbol{A}^l) \tag{4}$$

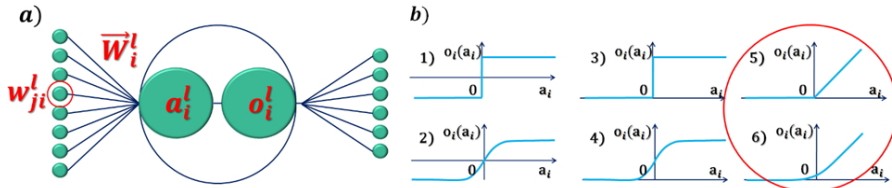

Figure 1: (a) A formal neuron of layer $l$: weight vector $\mathbf{W}_i^l$, linear activation $a_i^l$, and non-linear output activation $o_i^l$; (b) Common types of functions $\varphi(a_i^l)$, with 5—ReLU and 6—SoftPlus.

## 3 GRADIENT COMPUTATION USING THE BPE METHOD

When using the BPE method, the quantity that propagates backward through the network is not the "error" itself but the partial derivative of the loss function $E$ with respect to the values of neural network variables $a_i^l$ and $o_i^l$, denoted as $\frac{\partial E}{\partial a_i^l}$ and $\frac{\partial E}{\partial o_i^l}$. Let us define $\delta_i^l = \frac{\partial E}{\partial a_i^l}$. Based on this quantity $\delta_i^l$ and the output activity $o_j^{(l-1)}$, the weight increments $\Delta w_{ji}^l$—the changes in the components of the input weight vector $\mathbf{W}_i^l = \{w_{ji}^l\}$—are computed. Since:

$$\frac{\partial E}{\partial w_{ji}^l} = \frac{\partial E}{\partial a_i^l} \cdot \frac{\partial a_i^l}{\partial w_{ji}^l} = \delta_i^l o_j^{(l-1)}, \quad \text{since} \quad \frac{\partial a_i^l}{\partial w_{ji}^l} = o_j^{(l-1)}, \tag{5}$$

we obtain:

$$\Delta w_{ji}^l = -\alpha \frac{\partial E}{\partial w_{ji}^l} = -\alpha \delta_i^l o_j^{(l-1)}, \quad 0 < \alpha \ll 1. \tag{6}$$

In vector-matrix form, the change in the input weight matrix $W^l$ of layer $l$ is expressed as the outer product of vectors:

$$\Delta W^l = -\alpha \boldsymbol{\delta}^l (\boldsymbol{O}^{(l-1)})^T, \quad \Delta W^l = \{\Delta w_{ji}^l = -\alpha \delta_i^l o_j^{(l-1)}\}. \tag{7}$$

where $\boldsymbol{\delta}^l = \{\delta_i^l\}$ is a column vector and $(\boldsymbol{O}^{(l-1)})^T = \{o_j^{(l-1)}\}$ is a row vector. All weight updates in the input weight matrix $W^l$ follow the negative gradient of the loss function $E$ and are proportional to the "errors" $\delta_i^l$ (see Podoprosvetov et al. (2024) for further details).

## 4 OPTIMIZATION OF TRANSFORMATIONS IN NEURONS

Neural network parameter optimization aims to construct an approximation that transforms input signals into output signals with maximum accuracy. When solving complex recognition and generation tasks, the exact form of the optimal transformation function is typically unknown both before and after training. The accuracy of the approximation can only be indirectly assessed based on how well the training objectives are met.

To study this problem, it is preferable to examine simpler tasks where the reference transformation function is known and precisely formalized. One example is neural network approximation of transformations defined explicitly by analytical functions. While such tasks lack practical applications (since exact functional descriptions are more efficient), they serve as useful abstract models for studying adaptive processes. These models allow precise identification of approximation errors and investigation of their causes and possible mitigation strategies.

Neural network training involves adjusting the parameters of its constituent neurons. The back-propagation algorithm (BPE), which implements the idea of gradient descent, is currently the main optimization method. All input weights of neurons are adjusted, and although the bias parameter

(threshold, bias) has a different nature, it is conventionally described as a weight connected to a unit activation element and trained using the same rules as other weights.

## 5 TUNING VECTOR-MATRIX PARAMETERS OF A NEURAL NETWORK

The optimization of connection vectors $\boldsymbol{W}_i^l$, which form the rows of the weight matrices $W^l$, follows the BPE method (Equation 2) and is analogous to linear regression construction.

Let $y_i$ represent the ideal linear regression of the reference nonlinear transformation being learned by the neural network. Equations (1) (excluding the nonlinear part) define a hyperplane specified by the vector $\boldsymbol{W}_i^l$. The slope of this hyperplane along the component $o_j^{(l-1)}$ determines the weight $w_{ji}^l$, and the intersection with the $a_i^l$ axis (when all $o_j^{(l-1)} = 0$, except for $j = 0$) defines $w_{0i}^l$. According to the weight update rules (Equation 2), in batch training with $N_b$ training samples, the weight updates are given by:

$$\Delta w_{0i}^l = -\alpha \sum_{k=1}^{N_b} \delta_{ik}^l, \quad \Delta w_{ji}^l = -\alpha \sum_{k=1}^{N_b} (\delta_{ik}^l o_j^{(l-1)}), \quad k = 1, \ldots, N_b. \tag{8}$$

This leads to a gradual reduction in parameter updates to zero.

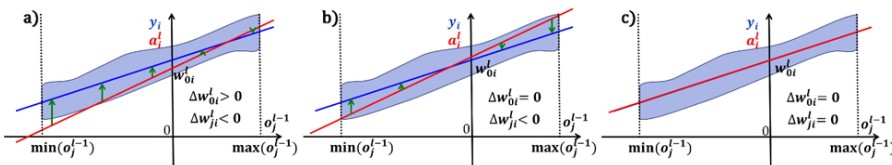

Figure 2: Sequential reduction of the parameter increments $w_{0i}^l$ and $w_{ji}^l$ to zero (3)

The adjustment of all parameters $w_{ji}^l$ in parallel using cumulative updates (Equation 3) results in a hyperplane approximating the training data as closely as possible, equivalent to constructing a linear regression model. However, if all network neurons were adjusted this way without local non-linearities, a single hidden-layer neuron would suffice for linear regression. Adding more neurons would merely replace one hyperplane with a sum of several, increasing computational complexity without improving the final result. Replacing linear transformations with nonlinear ones introduces new properties, not all of which (as shown in Section 7) are desirable.

High-dimensional activation spaces of hidden layers exhibit many useful properties, one of the most important being the presence of a large number of mutually perpendicular directions. These directions allow for the independent tuning of various network properties by nullifying the scalar products of vectors. This topic was previously discussed in Ramachandran et al. (2017), and we continue to develop it further. However, there are also other ways to partition the training process, which is the focus of this work.

## 6 SIMPLE MODEL FOR TUNING SCALAR NONLINEAR PROPERTIES

According to Equations (1) and (2), the argument of the functions $a_i^l$ and $\Delta w_{ji}^l$ is the output activity vector $\boldsymbol{O}^{(l-1)}$ of the previous layer $l-1$. In the high-dimensional state space of $\boldsymbol{O}^{(l-1)}$ with dimension $N^{(l-1)}$, identifying and understanding the influence of $\boldsymbol{O}^{(l-1)}$ on the forms of $a_i^l$ and $\Delta w_{ji}^l$ is complex. The problem can be simplified by decomposing the vector $\boldsymbol{O}^{(l-1)}$ into two components: one parallel and one perpendicular to the weight vector $\boldsymbol{W}_i^l$, as follows:

$$\boldsymbol{O}^{(l-1)} = \boldsymbol{O}_{\parallel}^{(l-1)} + \boldsymbol{O}_{\perp}^{(l-1)}. \tag{9}$$

We are only interested in the component $\boldsymbol{O}_{\parallel}^{(l-1)}$, which is the projection of $\boldsymbol{O}^{(l-1)}$ onto $\boldsymbol{W}_i^l$:

$$\boldsymbol{O}_{\parallel}^{(l-1)} = \boldsymbol{W}_i^l \frac{(\boldsymbol{W}_i^l \cdot \boldsymbol{O}^{(l-1)})}{(\boldsymbol{W}_i^l \cdot \boldsymbol{W}_i^l)}. \tag{10}$$

Since the activation function is given by:

$$a_i^l = \boldsymbol{W}_i^l \cdot \boldsymbol{O}^{(l-1)} = \boldsymbol{W}_i^l \cdot (\boldsymbol{O}_{\parallel}^{(l-1)} + \boldsymbol{O}_{\perp}^{(l-1)}) = \boldsymbol{W}_i^l \cdot \boldsymbol{O}_{\parallel}^{(l-1)}, \tag{11}$$

where the scalar product of perpendicular vectors $\boldsymbol{W}_i^l$ and $\boldsymbol{O}_{\perp}^{(l-1)}$ is zero. This reduction allows the analysis of nonlinear properties tuning to be reduced to a function of a single variable $u_i^l$, related to $a_i^l$ as follows:

$$a_i^l = p_i^l u_i^l + q_i^l, \tag{12}$$

where $p_i^l$ is a coefficient determined by $\boldsymbol{W}_i^l$, and $q_i^l = w_{0i}^l$ is the bias parameter. This one-dimensional model, though simplified, provides insights into the optimization process.

## 7 CAUSES OF LOCAL MINIMA IN OPTIMIZATION

The simplified one-dimensional model facilitates the analysis of neuron parameter tuning using the BPE method. Consider a simple case of approximating a ReLU-type function (shown in blue in Figure 3) by tuning the weight vector $\boldsymbol{W}_i^l$ of a neuron implementing a ReLU transformation. Ideally, weight adjustments should lead to near-zero approximation error. However, when using BPE, this is not always achieved (the red response in Figure 3).

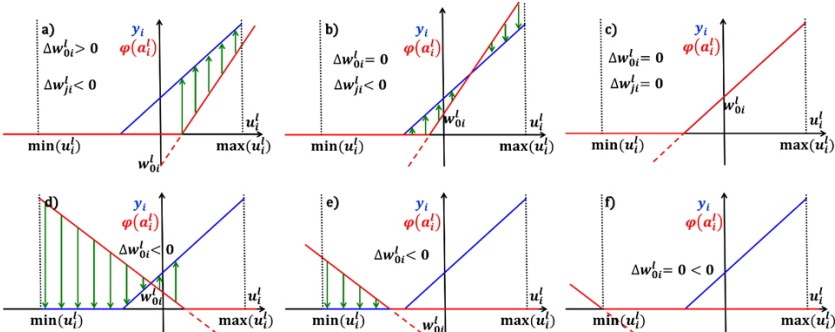

Figure 3: Gradient descent always reduces the batch sum of $\delta_{ik}^l$ (3), but does not always yield a good approximation of the target transformation.

If the growth direction of the linear activation matches that of the target function $y_i$ (Figure 3a), the shift parameter $w_{0i}^l$ is adjusted first (Figure 3b), quickly zeroing out $\sum \delta_{ik}^l$. Then, the other weights $w_{ji}^l$ are tuned more slowly (Figure 3c), until the sum of $\delta_{ik}^l o_j^{(l-1)}$ is minimized, leading to an optimal approximation.

If the growth direction of the linear activation is opposite to that of $y_i$ (Figure 3d), gradient descent still operates, but instead of adjusting signs, it minimizes $\sum \delta_{ik}^l$ by gradually deactivating the neuron (Figure 3f). Where $\phi'(a_i^l) = 0$, the values of $\delta_{ik}^l$ are also zero. This results in neuron dropout. Studies Atanov et al. (2019) indicate that this can affect over 90% of neurons.

To address this, ReLU is replaced with functions such as LeakyReLU or Swish Ramachandran et al. (2017), which do not have large zero-gradient regions. This allows "deactivated" neurons to resume operation, but the reconfiguration process is slow due to near-total (though not absolute, as in ReLU) deactivation. This causes non-monotonic distance reduction between initial and final parameter configurations, with growth phases. Our research aims to eliminate such failures and provide theoretical justification for our proposed transformation model. Our findings have so far

been validated only on exploratory tasks but suggest that our transformation structure could improve benchmark accuracy beyond the 0.9% gain achieved by Swish Ramachandran et al. (2017).

Another important aspect of our research is resource allocation across neural layers within the input state subspaces $\boldsymbol{O}^{(l-1)}$. Ignoring this issue leads to over-allocation in "successful" transformation regions while under-allocating for "difficult" areas. Distributing resources proportionally to transformation complexity, along with eliminating training failures, will accelerate network tuning and achieve deeper approximation minima. As with any scientific problem, progress relies on distinguishing optimization objectives and selecting targeted methods to achieve them.

# 8 DECOMPOSITION OF NEURAL NETWORK OPTIMIZATION TASKS INTO SPECIALIZED ALGORITHMS

Although the BPE method allows for tuning not only vector parameters but also other neural network parameters, the analysis above shows that it is highly likely to lead the optimization process to local minima. A possible way to overcome these shortcomings of BPE is to decompose the optimization process into separate subtasks and solve them using dedicated specialized algorithms.

Analysis identifies several subtasks in the optimization process:

1. Adjusting the bias values $w_{0i}^l$ of linear activation for all neurons in the regression task;

2. Selecting optimal directions and magnitudes for weight vectors $\vec{W}_i^l$ in the regression task;

3. Distributing the directions of $\vec{W}_i^l$ vectors in the state space of $\vec{O}^{(l-1)}$;

4. Adjusting the biases of nonlinear functions $\varphi(a_i^l)$;

5. Tuning the slope of the ReLU function;

6. Adjusting the curvature of the SoftPlus function;

7. Compensating for the effect of changes in weight matrices $\Delta W^l$ in previous layers;

8. Normalizing the activations propagating through the network ($\vec{A}^l$);

9. Normalizing the backpropagated error ($\vec{\delta}^l$);

10. Orthogonalizing the activity vectors of layers ($\vec{A}^l$).

The first two points in this list are effectively handled by the BPE method. To improve the efficiency of the remaining subtasks, several specialized algorithms will be described below. For instance, the third algorithm in the list is proposed to be implemented based on the self-organizing map (SOM) algorithm Marfo & Przybyła-Kasperek (2022), while points 4-6 involve the use of specialized algorithms based on the accumulation of statistical expectations of various nonlinear transformations. Algorithms for points 7-9 were previously discussed in Czarnowski (2022).

# 9 MODEL OF A "DECOMPOSED" FORMAL NEURON

In this context, the problem of "decomposed" approximation is considered as an independent adjustment of biases for nonlinear functions $\varphi(a_i^l)$, tuning of "angles" for ReLU and SoftPlus functions, and setting the curvature parameters of the SoftPlus function for all neurons in a layer. We consider the operation of a single neuron, which differs from others only in the values of its parameters.

Decomposing the training process into different learning algorithms is not aimed at improving the generality of the BPE method but at addressing the issues outlined in point 7, which cause slow training and convergence to shallow minima. Importantly, in the proposed configuration, the derivative of the nonlinearity with respect to its argument not only almost never approaches zero but also practically never tends towards it.

Distinctive features of the proposed transformation include:

- Individual nonlinearity parameters for each neuron: shift $q_i^l$ and slope $c_i^l$, which linearly define the argument $v_i^l$ of the nonlinear transformation $\varphi$, and parameter $d_i^l$, which determines (for a given $c_i^l$) the magnitude of derivatives $\varphi(v_i^l)$;

- Symmetry of the function $\varphi(v_i^l)$ relative to zero argument, achieved by adding linear terms to $2\max(v_i^l, 0)$ or $2\ln(1 + e^{v_i^l})$.

The nonlinear transformation $o_i^l(a_i^l)$ is defined by the following equations:

$$o_i^l = a_i^l + \varphi(v_i^l); \tag{13}$$

$$v_i^l = c_i^l(a_i^l + q_i^l); \tag{14}$$

$$\varphi(v_i^l) = d_i^l(2\max(v_i^l, 0) - v_i^l) \quad \text{(ReLU analog)} \tag{15}$$

$$\varphi(v_i^l) = d_i^l(2\ln(1 + e^{v_i^l}) - 2\ln 2 - v_i^l) \quad \text{(SoftPlus analog)} \tag{16}$$

The internal transformation parameter $v_i^l$ is linearly related to the external parameter $u_i^l$ mentioned earlier, but they are not equal. The interaction scheme of variables according to (5) is shown in Figure 4.

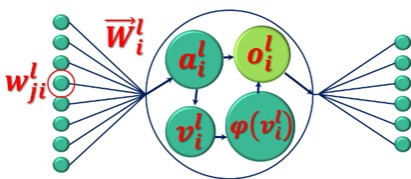

Figure 4: Structure of the "decomposed" formal neuron.

To tune the ReLU analog, it is sufficient to use only one of the two parameters $c_i^l$ and $d_i^l$ (e.g., setting $c_i^l = 1$), since changing either results in the same change in angle $\theta_i^l$ (see Figure 5a). However, both parameters are presented in equations (5) because, for the ReLU analog, it is easier to understand that changing the angle between asymptotes requires modifying the product $c_i^l d_i^l$. Conversely, if both parameters are changed while keeping their product constant, the angle remains unchanged (including for the SoftPlus analog).

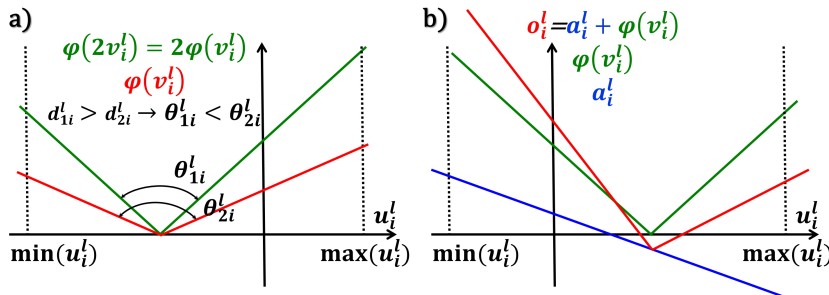

Figure 5: Properties of nonlinear transformations, ReLU and SoftPlus analogs (5).

As the product $c_i^l d_i^l$ decreases to zero, the nonlinearity $\varphi(v_i^l)$ degenerates into a straight line. Further reduction (into negative values) results in a sign change for both curvature and angle $\theta_i^l$. Since the functions $\max(v_i^l, 0)$ and $\ln(1 + e^{v_i^l})$ are always non-negative and symmetric about zero argument, changing the argument's sign does not affect their values. This means that to reverse the direction of angle $\theta_i^l$ (upward or downward), it suffices to change the sign of $d_i^l$ while keeping $c_i^l$ strictly positive (enforcing $c_i^l > 0$ programmatically).

## 10 ADJUSTMENT OF NONLINEARITY OFFSET AND ANGLE BETWEEN ASYMPTOTES

As follows from equations (5) and the graphs in Fig. 5, the adjustment of these parameters is carried out by adaptively changing the individual values of the coefficients $q_i^l$, $c_i^l$, and $d_i^l$ for each neuron. It

is crucial that the adjustment of these parameters always leads to a reduction in total error, similar to gradient descent in BPE, but with less tendency to settle in local minima. To achieve this, we utilize BPE-based ideas separately for the local parts of the nonlinear transformation, specifically to the left and right of the zero argument (scalar) of the nonlinear transformations (5).

The adjustment of scalar parameters is performed in parallel with the vector parameters of each neuron, ensuring that the equalities (3) are approximately maintained. This means that, overall, for each neuron, the sum:

$$\sum_{k=1}^{N_b} \delta_{ik}^l \approx 0, \tag{17}$$

but if divided into two sums, to the left and right of the point $u_i^l$ such that $v_i^l(u_i^l) = 0$, they can deviate significantly from zero:

$$\sum_{k=1}^{N_b} (\delta_{ik}^l)^{\text{left}} \approx -\sum_{k=1}^{N_b} (\delta_{ik}^l)^{\text{right}}, \tag{18}$$

$$\sum_{k=1}^{N_b} \left(\delta_{ik}^l o_j^{l-1}\right)^{\text{left}} \approx -\sum_{k=1}^{N_b} \left(\delta_{ik}^l o_j^{l-1}\right)^{\text{right}}. \tag{19}$$

To improve approximation accuracy, it is necessary for each of the semi-sums in (6) to also approach zero. However, neither BPE nor equality (6) ensure this. Applying a BPE-like approach separately to each semi-sum reduces the approximation error. Since equations (5) differ from (1) and achieving the desired result requires modifying the parameters $q_i^l$ and $d_i^l$ (and for SoftPlus, also $c_i^l$), the sums from (6) should not be linked to increments of vector components $\mathbf{W}_i^l$ as in (3), but rather to these parameters directly.

Since the equality of sums in (6) is approximate (and assumes that BPE has already sufficiently tuned $\mathbf{W}_i^l$ so that their increments in (3) are negligibly small—an assumption that may not hold, especially in early training), it is preferable to take into account their signs and magnitudes. The coefficients for the linear components $o_i^l = a_i^l + \phi(v_i^l)$ for the left and right parts behave differently under small changes in $q_i^l$ (Fig. 6a). When linear coefficients have the same sign, output activity changes in opposite directions; when they have different signs, the magnitude of changes differs. This affects the sums in (6), violating their approximate equality, but this discrepancy is compensated by adjusting $\mathbf{W}_i^l$ using BPE.

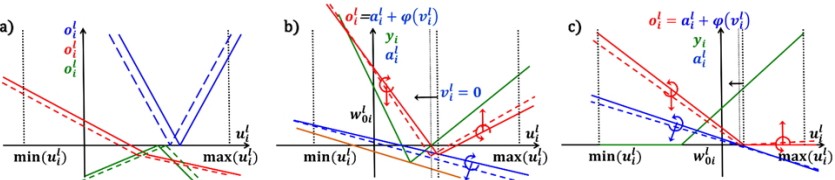

Figure 6: Adjustment of parameters $q_i^l$, $d_i^l$.

The linear coefficients $(r_i^l)^{\text{left}}$ and $(r_i^l)^{\text{right}}$ are formed according to (5) as algebraic sums of the coefficients $p_i^l = \|\mathbf{W}_i^l\|$ and $d_i^l$:

$$(r_i^l)^{\text{left}} = p_i^l - d_i^l, \quad (r_i^l)^{\text{right}} = p_i^l + d_i^l. \tag{20}$$

Increasing the offset parameter $q_i^l$ by a small amount $\Delta q_i^l$ shifts the point $v_i^l(u_i^l) = 0$ to the left and changes each $\delta_{ik}^l$ in the sum $\sum_{k=1}^{N_b}(\delta_{ik}^l)^{\text{right}}$ by $(r_i^l)^{\text{right}}\Delta q_i^l$. The total change in the sum is $(n_i^l)^{\text{right}}(r_i^l)^{\text{right}}\Delta q_i^l$, where $(n_i^l)^{\text{right}}$ is the number of terms in the sum. Similarly, for $(\delta_{ik}^l)^{\text{left}}$, we get:

$$(n_i^l)^{\text{right}}(r_i^l)^{\text{right}}\Delta q_i^l = -\gamma_1 \sum_{k=1}^{N_b}(\delta_{ik}^l)^{\text{right}}, \tag{21}$$

which gives:

$$\Delta q_i^l = \frac{-\gamma_1 \sum_{k=1}^{N_b} (\delta_{ik}^l)^{\text{right}}}{(n_i^l)^{\text{right}} (r_i^l)^{\text{right}}}.$$

(22)

A similar expression can be derived for $(\Delta q_i^l)^{\text{left}}$, but the shifts may not be equal and can even be in opposite directions. To ensure higher approximation accuracy, a weighted average value of $\Delta q_i^l$ is chosen:

$$\Delta q_i^l = \frac{(\Delta q_i^l)^{\text{left}} \left| \sum_{k=1}^{N_b} (\delta_{ik}^l)^{\text{left}} \right| + (\Delta q_i^l)^{\text{right}} \left| \sum_{k=1}^{N_b} (\delta_{ik}^l)^{\text{right}} \right|}{\left| \sum_{k=1}^{N_b} (\delta_{ik}^l)^{\text{left}} \right| + \left| \sum_{k=1}^{N_b} (\delta_{ik}^l)^{\text{right}} \right|}.$$

(23)

Similarly, a weighted value for $\Delta d_i^l$ is calculated, which determines the change in angle $\theta_i^l$. The complexity in defining $\Delta d_i^l$ arises from the dependence of sum variations on the "arm lengths" of the sides. To determine the mean "arm length," all values $(a_i^l - q_i^l)$ to the right of $v_i^l = 0$ and $(q_i^l - a_i^l)$ to the left are summed. If using a more complex nonlinearity than ReLU, these sums must be multiplied by $c_i^l$. The total length sum, multiplied by $\Delta d_i^l$, gives:

$$\Delta d_i^l = \frac{-\gamma \sum_{k=1}^{N_b} (\delta_{ik}^l o_j^{l-1})^{\text{right}}}{\sum_{k=1}^{N_b} (a_i^l - q_i^l)^{\text{right}}}.$$

(24)

A weighted total $\Delta d_i^l$ is computed similarly to (9). The effects of applying formulas (7)–(10) are illustrated in Fig. 6b and 6c. Figure 6b shows an arbitrary example of shifting the "separated" ReLU function to the target position through shifts (due to changes in $q_i^l$ and $w_{0i}^l$) and rotations (due to changes in $d_i^l$ and $\vec{W}_i^l$) of its components. Figure 6c applies the training of the "separated" ReLU function to the case previously considered in Figure 3d-f. Unlike the standard ReLU function, the "separated" version does not encounter adaptation issues.

## 11 Curvature Adjustment of the SoftPlus Function

When approximating smooth functions, an important source of deviation from the reference function is the presence of derivative discontinuities in the ReLU function. Even a simple replacement of ReLU with Swish Zheng et al. (2015) improves approximation across a wide range of tasks. The "separated" SoftPlus function also has no derivative discontinuities and, moreover, allows tuning of the "curvature" parameter to match the properties of the approximated function. This adjustment is performed individually (but, like all other neural network algorithms, in a mass manner) and serves as an additional fine-tuning feature on top of all the previously described advantages of the "separated" ReLU.

## 12 Compensation for Weight Matrix Adjustments in Earlier Layer

All the aforementioned modifications, aimed at separating the optimization methods of neural network transformations, influence the direction of parameter updates, which can significantly deviate from the direction of the negative gradient of the loss function $E$ with respect to the parameters. While such deviations may be beneficial in hidden layers, in the output layer, it is preferable not to deviate from the optimal approximation direction. This can be ensured by not only avoiding additional algorithms when tuning the final weight matrix but also compensating for the influence of changes in matrices $\Delta W^l$ in previous layers. The algorithm for computing compensatory additions to the vectors $\vec{\delta}^l$ is described in Podoprosvetov et al. (2024).

## 13 Parameter Normalization

Stable operation and training of a neural network can be achieved by ensuring smooth propagation of activity $\vec{A}^l$ and "error" $\vec{\delta}^l$ through the network, avoiding sharp spikes or dampening. The ability

to compensate for the influence of additional algorithms on the training of subsequent layers allows normalizing activities $\vec{A}^l$ by introducing an algorithm for modifying weight matrices $W^l$, while normalizing $\vec{\delta}^l$ is embedded in the BPE method. The use of compensation ensures that the additional learning algorithms (including all those described above) do not distort the gradient descent direction in the last hidden layer, where only the BPE method is applied.

## 14 Orthogonalization of Activity Vectors in Neural Network Layers

Another important idea in "separating" the optimization of neural network transformations is the approach outlined in Podoprosvetov et al. (2024), which utilizes a key property of high-dimensional vector spaces—the presence of a vast number of mutually orthogonal directions. This approach not only normalizes the magnitude of vectors $\vec{A}^l$, but also adjusts the matrices $W^l$ to maximize the distribution of $\vec{A}^l$ across its subspace of states. This significantly reduces the dependency between learning parameters for different input signals, leading to increased learning speed and accuracy.

## 15 Results

When applied together with the normalization, orthogonalization, and load distribution methods described in previous reports, the optimization algorithms for nonlinear transformation parameters presented here enable a reduction in training time (depending on the type of transformation) by a factor of 3 to 10 and improve approximation accuracy by 1.5 to 2 orders of magnitude for ReLU and up to 3 orders of magnitude for SoftPlus. These results were obtained through the modeling of neural network approximation tasks for analytically defined vector functions. Improving the accuracy of approximating smooth functions is critical for many applications, such as robotics, where precise motion description and prediction are essential. Moreover, even in recognition tasks and similar applications, where exact representation of transformations may not seem strictly necessary, increasing approximation accuracy can enhance the optimization of transformations.

## 16 Conclusions

The success of solving complex "intelligent" tasks using powerful systems based on LLM training and other modern approaches should not create the impression that all neural network parameter tuning algorithms are fully understood and that no new developments can emerge in this field. On the contrary, vector-matrix analysis of various aspects of neural network transformations presents a vast area for research and promising developments, as more efficient algorithms scale better to large and complex tasks.

This work focused on the potential of separating optimization tasks in neural network parameter tuning processes. Even in a relatively simple piecewise-linear neural network approximation based on the ReLU function, it is possible to separate the training of vector-linear and scalar-nonlinear transformations. Moreover, since most nonlinear functions used in neural networks have linear asymptotes, the obtained results can be extended to them, primarily to SoftPlus, as a smooth analog of ReLU. Furthermore, the use of smooth, derivative-continuous functions enables finer-tuned approximation adjustments by aligning the curvature of the reference and neural transformation functions. However, fine-tuning methods are effective only if coarser optimization algorithms do not encounter adaptation problems.

Beyond improving individual aspects of neural network algorithms, this work is valuable as an example of using vector-matrix analysis to study the properties of neural network data processing. Vector-matrix analysis provides a deeper understanding of the transformations performed and suggests ways to accelerate and improve the accuracy of neural network approximation processes.

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
