# OpenReview forum: "Separate Adjustment of Linear and Nonlinear Parameters in Neural Network Training"
_mathai.club/MathAI/2025/Conference — MathAI 2025 Oral_

### Official Review · Reviewer_s36C · 2025-02-25
**This paper proposes model of a ”separated” formal neuron, based on the idea that splitting the parameters of nonlinear activation functions into individual and customizable components can significantly improve the training process of neural networks. By  applying several methods and optimization algorithms for nonlinear transformation parameters authors show that they enable a reduction in training time on approximation tasks for analytically defined vector functions.**

**Rating:** 7
**Confidence:** 4

**Review:**

Detailed Review. Quality & Clarity The paper is well-organized and give clear description of approach, including the decomposition of optimization process into separate subtasks and solve them using dedicated specialized algorithms. For studying adaptive processes author use neural network approximation of transformations defined explicitly by analytical functions. These models allow precise identification of approximation errors, but can‘t be real evaluation of model ( for example due to absence of information errors).
Originality and Significance. The study is aimed to important sphere of the investigation of the backpropagation error (BPE) method in neural network training, particularly its tendency to converge to suboptimal local minima. Approach, based on dividing optimization in several steps is new. The results is rather theoretical, but can be used in solving practical tasks of network training.
 Pros and Cons Pros. Investigate the fundamental task of neural network training process optimization. But digits in experiment of reduction in training time and improvement approximation accuracy, achieved by the authors, need clear explanation.

---

### Official Review · Reviewer_a1N7 · 2025-02-27
**Well-structured paper with a novel approach to neural network optimization, showing promising theoretical results but needing more empirical validation**

**Rating:** 7
**Confidence:** 4

**Review:**

The paper is well-structured and clearly presents the proposed approach, which involves separating the optimization of linear and nonlinear parameters in neural network training. The authors provide a detailed explanation of the methodology and demonstrate its advantages through function approximation experiments. The mathematical formulation based on vector-matrix analysis is well-explained, and the results are presented with relevant comparisons. However, while the theoretical foundation is solid, additional clarification on the practical applicability of the method to real-world deep learning tasks would strengthen the paper.


## Strengths

- Introduces a novel approach to neural network training by decoupling linear and nonlinear parameter optimization.
- Theoretical analysis based on vector-matrix formulation is rigorous and well-presented.
- Experimental results show significant improvements in training speed and accuracy, particularly with SoftPlus activation.


## Weaknesses
- The practical applicability of the method beyond function approximation tasks is not fully explored.
- Reported improvements in accuracy and training time require more detailed justification and comparison with state-of-the-art techniques.

## Final Evaluation

The paper presents a valuable contribution to neural network training optimization with a well-structured theoretical approach. While the results are promising, further clarification on the real-world applicability and empirical validation would enhance the impact of the work.

---

### Official Review · Reviewer_LWdf · 2025-02-27
**The review of SEPARATE ADJUSTMENT OF LINEAR AND NONLINEAR PARAMETERS IN NEURAL NETWORK TRAINING**

**Rating:** 7
**Confidence:** 4

**Review:**

The paper presents broad and detailed investigation of methods for decomposition of neural network optimisation in contrast to backpropogation error. The authors explicitly show negative effects appearing in neural network optimisation and approaches to deal with them with demonstrative special cases and theoretical proof for more general cases.

The authors claim improvement of accuracy and reduction of training time in previous reports, but they are not referenced and it is not clear what are results compared against.

The paper has couple minor technical flaws in missing references in sections 11 and 12.

Overall, the paper offers a clear depiction of approaches to decomposition of neural network optimisation, however presenting estimation of the practical effect on more complex examples would be beneficial.

---

### Official Review · Reviewer_tR2F · 2025-02-28
**A paper analyzing gradient flow through the layers of a neural network and proposing distinct methods to optimize different kinds of model parameters**

**Rating:** 7
**Confidence:** 4

**Review:**

### Overview

While neural networks are known to be universal approximators and modern systems such as LLMs show great performance in „intelligent“ tasks, we still don’t understand the NN training process deeply. But the authors make a step towards this direction dissecting the NNs and providing insights into the learning process. The paper introduces an alternative neuron model and adjust learning process with specialized methods for each type of adjustment to impose properties that are considered good for gradient flow. Hence neural network learning isn’t treated as a blackbox optimization problem. This is a relevant research topic since new NN architectures and insights in learning process are in the focus of attention nowadays.

### Weaknesses

The experimental section of a moderate size and lacks important details. Only all the adjustments together are tested preventing the evaluation of the seemingly independent improvements. If it’s impossible, this fact is worth mentioning in the discussion. Also, the actual metric values aren’t provided, only relative improvements. Optionally, there could be experiments on well-known datasets where the optimization part of this method is applied to the state-of-the-art models.

The review of the previous work pays little attention to the existing effort to learning activation functions (such as [1]) and adaptive optimization (such as [2]).

[1]  KAN: Kolmogorov-Arnold Networks (Liu et al.)

[2]  Learning to Optimize (Ke Li, Jitendra Malik)

### Typesetting

289 — this list is referred by index and should be turned into a numberer list for clarity

361 — the OX axis of right graph of diagram doesn’t contain indices of u

480, 496 — references aren’t rendered

It is suggested to make Sections 10-14 subsections of a bigger one since they are closely related.

---

### Decision · Program_Chairs · 2025-03-08

**Decision:**

Accept (Oral)

**Comment:**

Your article has been accepted and you can make a presentation on the article. All articles will be sorted by rating and within the available conference places one author from each article will be invited. If there are not enough places, then you will either have the opportunity to present remotely or come at your own expense!